# Patients' and carers' priorities for cancer research in Aotearoa/New Zealand

**Millie de Vries**[1☯], **Tiria Stewart**[2☯], **Theona Ireton**[3], **Karen Keelan**[4], **Jennifer Jordan**[5,6], **Bridget A. Robinson**[1,7], **Gabi U. Dachs**[1]*

1 Mackenzie Cancer Research Group, Department of Pathology and Biomedical Science, University of Otago, Christchurch, New Zealand (NZ), 2 Te Pūtahi Mātai Toto o Te Waipounamu, Christchurch Hospital, Christchurch, NZ (Nga Puhi, Ngāti Porou), 3 Māori Health Services, Christchurch Hospital, Te Whatu Ora, Waitaha/Canterbury, Christchurch, NZ (Ngā Wairiki, Ngāti Porou), 4 Te Aho o Te Kahu–Cancer Control Agency, Ministry of Health, NZ (Ngāti Porou), 5 Psychological Medicine, University of Otago, Christchurch, NZ, 6 Specialist Mental Health Service Clinical Research Unit, Te Whatu Ora, Waitaha/Canterbury, Christchurch, NZ, 7 Canterbury Regional Cancer and Haematology Service, Te Whatu Ora, Waitaha/Canterbury, Christchurch, NZ

☯ These authors contributed equally to this work.
* gabi.dachs@otago.ac.nz

**Data Availability Statement:** All relevant data are within the paper and its Supporting Information files.

**Funding:** Part-funding was obtained from the Cancer Society NZ (MV) and the Mackenzie

## Abstract

### Background

Discrepancies have been reported between what is being researched, and what patients/ families deem important to be investigated. Our aim was to understand research priorities for those who live with cancer in Aotearoa/New Zealand, with emphasis on Māori.

### Methods

Adult outpatients with cancer and their whānau/family completed a survey (demographics, selecting keywords, free-text comments) at Christchurch hospital. Quantitative and qualitative data were evaluated using standard statistical and thematic analyses, respectively.

### Results

We recruited 205 participants, including both tūroro/patients (n = 129) and their whānau/ family/carer (n = 76). Partnership with Māori health workers enabled greater recruitment of Māori participants (19%), compared to the proportion of Māori in Canterbury (9%). Cancer research was seen as a priority by 96% of participants. Priorities were similar between Māori and non-Māori participants, with the keywords 'Cancer screening', 'Quality of Life' and 'Development of new drugs' chosen most often. Free-text analysis identified three themes; 'Genetics and Prevention', 'Early Detection and Treatment', and 'Service Delivery', with some differences by ethnicity.

### Conclusions

Cancer research is a high priority for those living with cancer. In addition, participants want researchers to listen to their immediate and practical needs. These findings may inform future cancer research in Aotearoa.

Charitable Foundation (BAR, GUD). The remaining
authors received no specific funding for this work.
The funders had no role in study design, data
collection and analysis, decision to publish, or
preparation of the manuscript.

**Competing interests:** The authors have declared
that no competing interests exist.

## Māori terms and translation

Aotearoa (New Zealand)

he aha ō whakaaro (what are your thoughts)

hui (gathering)

mate pukupuku (cancer)

mokopuna (descendent)

Ōtautahi (Christchurch)

rongoā (traditional healing)

tāne (male)

te reo (Māori language)

Te Whatu Ora (weaving of wellness, Health New Zealand)

tikanga (methods, customary practices)

tūroro (patients) (alternative terms used: whānau affected by cancer or tangata whaiora
(person seeking health))

wahine (female)

Waitaha (Canterbury)

whakapapa (genealogy)

whānau ((extended) family, based on whakapapa, here also carer)

## Background

Cancer is a significant health burden in Aotearoa/New Zealand. Every year over 25,000 New
Zealanders find out that they have cancer, and over 9,000 die from their disease [1]. Patterns of
cancer registrations have changed over the years, with some cancers increasing (e.g. liver, pan-
creatic) and other cancers decreasing (e.g. stomach, lung) [1, 2]. Mortality has reduced signifi-
cantly for many cancer types over the last thirty years (data 1988 vs 2018: 165.2 vs 114.0 deaths
per 100,000) [1], due to reduction of risky behaviours (eg smoking), intensive screening and
improved treatments, but counterbalanced by an aging population and increased rates of some
comorbidities (eg obesity and diabetes) [3]. However, not all patients benefit, and ethnic dis-
parities are evident along the entire cancer continuum [2]. Māori are about 20% more likely to
develop cancer, and twice as likely to die from cancer than non-Māori [2].

The New Zealand Cancer Action Plan (Te Mahere mō te Mate Pukupuku o Aotearoa)
2019–2029 was developed to address these disparities and to ensure that all New Zealanders
experience equitable cancer outcomes [4]. Improvements in cancer survival are incremental,
and successful strategies are evidence-based and research-led. Cancer research in Aotearoa is
largely led by clinical and scientific researchers. Due to Aotearoa's small population size, clini-
cal research also often partners with international commercial clinical trials, but there are
issues with these internationally-driven clinical paradigms which are underpinned by assump-
tions such as that all genders and ethnicities will respond similarly to treatments, or have simi-
lar beliefs about treatment and research [5]. Yet we know little about the research priorities of
those who live with cancer, the tūroro/patients and their whānau/families. Under-representa-
tion of indigenous populations [6], such as Māori, in research is common and problematic, as
their views on research priorities are unknown.

Research priorities have been investigated in international studies [7–9], using a variety of
approaches and prioritisation methods. Formal methodologies have been developed to achieve

this, but none appear to be clearly advantageous for priority setting in Aotearoa [10]. One recent approach, the James Lind Alliance (JLA) method has been used for setting research priorities for the National Health Service in the United Kingdom [11]. The JLA approach uses a combination of surveys and workshops that involve patients, carers and health professionals to nominate the top unsolved research questions.

Using the JLA method, an analysis of the priorities of patients, clinicians and research communities has identified important mismatches [8]. The patient-clinician partnership prioritised education, training and service delivery, whereas both commercial and non-commercial trials prioritised drug development and other treatments [8].

Our study set out to determine the cancer research priorities of people living with cancer in Ōtautahi/Christchurch, Aotearoa. As limited data are available regarding cancer research priorities in Aotearoa [12, 13], we chose to adopt a simple survey format to gauge feedback from our community. Using structured and semi-structured questions, we aimed to determine the broad priorities of people living with cancer, how these may differ by ethnicity, sex and whether survey respondents were tūroro/patients or their whānau/family. We purposefully used Te Reo Māori in parts of the survey to enhance participation and improve inclusivity of Aotearoa's indigenous population.

## Methods

### Ethics and ethnicity

This study was approved by the University of Otago Human Ethics Committee (H20/150). Ethnicity was self-declared, referring to cultural identity, and can contain individuals with more than one ethnic affiliation. Ethnicity data was collected using the 2013 census questionnaire. We used a prioritised ethnic classification system: individuals are classified as Māori if Māori was self-reported as one of the ethnic groups in any ethnicity field. This is a high priority for health and research organisations in Aotearoa/NZ to address health inequities. Everyone else was grouped under the term non-Māori as a comparator. All collected data was deidentified and individuals are not identifiable.

### Recruitment

Participants included tūroro/patients with cancer and their whānau/family, friends or carers. Participants were recruited in-person by a medical student (MV or Māori health worker (TS) in the Oncology Unit or the South Island Bone Marrow Transplant Unit of Christchurch Hospital, respectively (Ōtautahi/Christchurch, Aotearoa/New Zealand), between November 2020 and July 2021. The two recruiters are specially trained to communicate with patients and were independent from the tūroro/patient's treatment team. Christchurch hospital is the main tertiary hospital on the South Island of Aotearoa/New Zealand. Participants were provided with an information sheet, and consented to completing a simple paper survey. Inclusion criteria were: aged $\geq$ 18 years, tūroro/patients with a confirmed diagnosis of cancer, and/or member/s of their whānau/family or friend/carer/support person; tūroro/patients and whānau/family consented independently. The only exclusion was the inability to provide informed consent.

### Survey and data collection

The simple 2-page survey "Mate Pukupuku: He aha ō whakaaro?" ("Cancer: What are your thoughts?") was created specifically for this study in consultation with Māori (KK), and following discussions with clinicians at Christchurch Hospital (BR) and scientists at the University

of Otago Christchurch (GD). A copy of the survey is available upon request from the corresponding author.

The survey comprised four sections, with section 1 collecting information on demographics (age, sex, ethnicity) and cancer type (for tūroro/patients only). In section 2 participants indicated priority rating of four broad research areas that broadly span the cancer continuum from cancer prevention, to diagnosis and prognosis of cancer, to reducing side effects of cancer treatment, and finally improving survival outcomes. Priorities were rated on a 4-point scale from 'very low' (= 1) to 'very high' (= 4). In section 3, participants were provided with a list of 44 cancer-related keywords, compiled from discussions with scientists and clinicians at the University of Otago Christchurch and Christchurch Hospital. Of these keywords, any number could be chosen as being important, with participants circling words to indicate their choice of research priorities. The final section was a free-text question "What else do you think is important for cancer researchers to study?", inviting any additional input.

Hard copy paper surveys were completed by participants usually during their clinical visit at Christchurch hospital. Participants could either fill the survey in themselves or had help from the medical student or Māori health worker. Participant survey data were anonymised and each participant was allocated a unique research code. De-identified answers were transcribed onto an Excel spreadsheet, and stored as password-protected electronic files on a secure server.

## Data analysis

This is a quantitative, cross-sectional study with an open-ended question for additional input. Quantitative data was analysed using descriptive statistics for categorical variables with GraphPad Prism 9. Normality of data was tested using the D'Agostino & Pearson test, followed by the Mann Whitney t-test (for non-parametric analysis) to compare between groups, with $p < 0.05$ being statistically significant. For qualitative data (free-text question), inductive thematic analysis and coding methods were used [14]. Codes identified from multiple readings of the raw data were grouped into categories independently by two researchers (MV, GD). Any differences were discussed and agreement reached. Categories were then synthesised into themes and sub-themes.

## Results

### Study cohort

We recruited 205 participants, including both tūroro/patients (n = 129) and their whānau/family/support (n = 76) (**Table 1**). The average age was 59.7 years for tūroro and 53.1 years for whānau, and 60% were wahine/female. Our partnership with Māori health workers enabled greater recruitment of Māori participants (19%) compared to the proportion of Māori in the region (9.4% [15]). The majority of participants categorised as non-Māori identified as NZ European (69%), then Samoan (2.4%) and European/British (2.4%), with the remainder consisting of ethnicities with fewer than five participants each (**Table 1**). As expected, the most common cancer types were breast, colorectal and prostate cancer [1]. Participants answered three types of questions for the survey: rating research priorities, choosing (most) important keyword from a list of cancer-related keywords, and adding further priorities as free text.

### Rating research areas

Participants rated four broad research areas from 'very low' to 'very high' priority (**Table 2**). The vast majority (96%) of participants rated research of all four research areas (*Cancer*

**Table 1. Characteristics of the cohort.**

| Demographics | | | | n = | % |
|---|---|---|---|---|---|
| *Cohort* | *total* | | | *205* | *100.0* |
| **Ethnicity** | Māori | | | 39 | 19.0 |
| | Non- Māori | | | 166 | 81.0 |
| | | *NZ European* | | *141* | *68.8* |
| | | *Samoan* | | *5* | *2.4* |
| | | *Europ/British* | | *5* | *2.4* |
| | | *Australian* | | *4* | *2.0* |
| | | *Other* | | *11* | *5.4* |
| **Age** | <50 years | | | 57 | 27.8 |
| | 50+ years | | | 147 | 71.7 |
| | not stated | | | 1 | 0.5 |
| **Sex** | Wahine/Female | | | 123 | 60.0 |
| | Tāne/Male | | | 81 | 39.5 |
| | Not stated | | | 1 | 0.5 |
| **Participants** | Tūroro/Patients | | | 129 | 62.9 |
| | Whānau/Family | | | 76 | 37.1 |
| **Cancer types** | *All* | | | *129* | *100.0* |
| | Breast | | | 29 | 22.5 |
| | Bowel | | | 16 | 12.4 |
| | Prostate | | | 14 | 10.9 |
| | Lung | | | 9 | 7.0 |
| | Lymphoma | | | 8 | 6.2 |
| | Melanoma | | | 8 | 6.2 |
| | Leukaemia | | | 6 | 4.7 |
| | other | | | 37 | 28.7 |
| | not stated | | | 2 | 1.6 |

*Prevention, Diagnosis/Prognosis of Cancer, Reducing Side Effects*, and *Improving Outcomes from Cancer Treatment*) as 'high' or 'very high'. The four research areas were ranked in order of *Improving Cancer Outcomes, Diagnosis/Prognosis of Cancer, Cancer Prevention* and *Reducing Side Effects*. When analysed by sub-groups, tūroro rated *Side Effects* and *Outcomes* significantly

**Table 2. Cancer research priorities according to ethnicity, sex or status.**

| | Category | n = | Cancer Prevention | Diagnosis/ prognosis | Side effects | Outcomes |
|---|---|---|---|---|---|---|
| **All** | **Total** | 205 | 3.68 | 3.73 | 3.50 | 3.75 |
| **Ethnicity** | **Māori** | 39 | 3.74 | 3.82 | 3.50 | 3.74 |
| | **non-Māori** | 166 | 3.66 | 3.70 | 3.49 | 3.76 |
| **Sex** | **Wahine/ Female** | 123 | 3.74* | 3.80** | 3.64*** | 3.82* |
| | **Tāne/Male** | 81 | 3.59 | 3.61 | 3.27 | 3.66 |
| **Status** | **Tūroro/ Patients** | 129 | 3.63 | 3.69 | 3.40* | 3.69* |
| | **Whānau/ Family** | 76 | 3.75 | 3.78 | 3.65 | 3.86 |

Average priorities scores are shown; priorities were rated from 'very low' (1) to 'very high' (4).

Mann Whitney comparing between sexes or between status

\* $p < 0.05$

\*\* $p < 0.01$

\*\*\* $p < 0.001$.

lower than whānau, whereas wahine rated all four areas significantly higher than tāne (**Table 2**).

## Cancer-related keywords

The second question provided participants with a list of 44 cancer-related keywords from which any number could be chosen as being important. For all participants together, a median of 12 keywords was chosen by participants (range 0–44), and 12% chose all 44 keywords. 'Cancer screening' was the top priority (72% of participants), followed by 'bowel cancer' and 'breast cancer' (57% and 56% respectively) and 'development of new drugs' (56%) (**Table 3**). The keywords 'equity' (20%), 'smoking' (24%) and 'alcohol' (28%) were rarely chosen.

We were interested to see whether these priorities (by keywords) were different between Māori and non-Māori participants, between wahine/females and tāne/males, or between tūroro/patients with cancer and their whānau/family (**Table 3**). Data for all keywords chosen by >50% of at least one subgroup are shown. When considering choices made by 60% or more of the participants, Māori participants chose 'quality of life', 'breast cancer' and 'Māori health', whereas non-Māori participants chose 'cancer screening', with the remaining keywords chosen by fewer than 60% of participants. Similarly, both wahine and tāne chose cancer screening as top priority (72% and 73%, respectively), followed for wahine by breast cancer (63%), bowel cancer (60%) and side effects (59%), and for tāne followed by development of new drugs (52%), inherited cancers (52%) and side effects (51%). Both tūroro and whānau chose 'cancer screening' as their top priority (74% and 68%, respectively). Aside from specific cancer types (bowel, breast and prostate cancer), it was interesting to see that patients more often chose 'new drug development' (57%) and 'cancer prevention' (56%), whereas non-patients more often chose 'quality of life' (59%) and 'reducing side effects' (55%) (**Table 3**).

**Table 3. Top chosen keywords from all participants and according to ethnicity, sex and status.**

|  | Cohort | | Māori | | non-Māori | | Wahine/Female | | Tāne/Male | | Tūroro/Patients | | Whānau/Family | |
|---|---|---|---|---|---|---|---|---|---|---|---|---|---|---|
|  | (n = 205) | | (n = 39) | | (n = 166) | | (n = 123) | | (n = 81) | | (n = 129) | | (n = 76) | |
|  | n = | % | n = | % | n = | % | n = | % | n = | % | n = | % | n = | % |
| **Quality of Life** | 111 | **54.1** | 25 | **64.1** | 86 | **51.8** | 66 | **53.7** | 35 | 42.7 | 66 | **51.2** | 45 | **59.2** |
| **Breast cancer** | 115 | **56.1** | 24 | **61.5** | 91 | **54.8** | 78 | **63.4** | 39 | 47.6 | 74 | **57.4** | 41 | **53.9** |
| **Māori Health** | 61 | 29.8 | 24 | **61.5** | 37 | 22.3 | 41 | 33.3 | 20 | 24.4 | 33 | 25.6 | 28 | 36.8 |
| **Cancer Screening** | 148 | **72.2** | 23 | **59.0** | 125 | **75.3** | 88 | **71.5** | 60 | **73.2** | 96 | **74.4** | 52 | **68.4** |
| **Inherited cancers** | 97 | 47.3 | 23 | **59.0** | 74 | 44.6 | 67 | **54.5** | 43 | **52.4** | 57 | 44.2 | 40 | **52.6** |
| **Development of new drugs** | 114 | **55.6** | 22 | **56.4** | 92 | **55.4** | 68 | **55.3** | 43 | **52.4** | 73 | **56.6** | 41 | **53.9** |
| **Mental health and cancer** | 85 | 41.5 | 22 | **56.4** | 63 | 38.0 | 57 | 46.3 | 32 | 39.0 | 48 | 37.2 | 37 | 48.7 |
| **Cancer prevention** | 111 | **54.1** | 21 | **53.8** | 90 | **54.2** | 71 | **57.7** | 40 | 48.8 | 72 | **55.8** | 39 | **51.3** |
| **Men's health** | 88 | 42.9 | 21 | **53.8** | 67 | 40.4 | 46 | 37.4 | 26 | 31.7 | 51 | 39.5 | 37 | 48.7 |
| **Bowel Cancer** | 117 | **57.1** | 20 | **51.3** | 97 | **58.4** | 74 | **60.2** | 41 | **50.0** | 76 | **58.9** | 41 | **53.9** |
| **Cancer genetics** | 99 | 48.3 | 20 | **51.3** | 79 | 47.6 | 65 | **52.8** | 34 | 41.5 | 64 | 49.6 | 35 | 46.1 |
| **Cannabis and cancer** | 76 | 37.1 | 20 | **51.3** | 56 | 33.7 | 52 | 42.3 | 30 | 36.6 | 42 | 32.6 | 34 | 44.7 |
| **Pain relief and cancer** | 99 | 48.3 | 19 | 48.7 | 80 | 48.2 | 63 | **51.2** | 35 | 42.7 | 64 | 49.6 | 35 | 46.1 |
| **Diet and cancer** | 98 | 47.8 | 18 | 46.2 | 80 | 48.2 | 63 | **51.2** | 36 | 43.9 | 58 | 45.0 | 40 | **52.6** |
| **Prostate cancer** | 101 | 49.3 | 17 | 43.6 | 84 | **50.6** | 51 | 41.5 | 28 | 34.1 | 59 | 45.7 | 42 | **55.3** |
| **Side effects of treatment** | 110 | **53.7** | 17 | 43.6 | 93 | **56.0** | 72 | **58.5** | 42 | **51.2** | 68 | **52.7** | 42 | **55.3** |

Note: Participants could choose 0–44 keywords. One participant did not state their sex. All keywords chosen by ≥50% of at least one subgroup are shown. Keywords are presented in order chosen by Māori participants. Keywords chosen by ≥50% of the subgroup are shown in bold.

## Free text answers

The third question of the survey invited participants to write down 'What else. . .', and was analysed qualitatively. From these free-text answers (n = 106/205 participants), three main themes were identified: '*Why did I get cancer*?' (genetics and prevention), '*Can you detect cancer earlier and treat it better*?' (early detection and treatment), and '*Can you look after our patients better*?' (service delivery), each with several subthemes. Examples of de-identified participant comments are shown below.

**Theme 1. Why did I get cancer?.** *Subtheme 1.1 Cancer risks and causes.* Numerous participants expressed their anguish at not knowing why they, themselves, or their whānau, developed cancer. Many tried to find a reason why it happened, possibly in order to prevent others getting cancer or to blame something. Some stated that they lived a healthy life and did everything right, yet they developed cancer.

> "*I still have no idea of what may be the cause of my cancer. It was heart breaking because I made the decision to look after myself by eating well and exercise. . .*" (CS137)

Putative environmental cancer risks were mentioned frequently, including *"nitrate in water"* (CS019), *"estrogen in rivers"* (CS025), *"hormone replacement therapy"* (CS041), *"cleaning products"* (CS140) and *"dental metals"* (CS123). Cancer incidence in Aotearoa as a whole or in specific locations and age groups, was deemed important to investigate (*"thirties aged people with bowel cancer..in Westport"* (CS003), *"why Canterbury NZ has the highest rate of bowel cancer/capita in the world"* (CS025)).

*Subtheme 1.2 Nutrition.* Nutrition and diet were mentioned as potential risk factors, but also as putative preventative measures (*"Food (what I can/can not have), diet"* (CS193), *"I am especially interested in learning more about diet"* (CS019). Participants provided very limited details for their answers.

*Subtheme 1.3 Familial cancer and genetics.* The inherited risk of developing cancer was seen as requiring further investigation, including specific genetic syndromes (*"Lynch syndrome"*). This was likely driven by a need to understand whether tūroro/patients inherited their cancer and the possible effect on their children and future generations. A mindset was apparent in older tūroro/patients, that genetic testing was helpful for the next generation, and less for themselves.

> "*To search up the link in family genes to see where it begins*" (CS188)

> "*DNA (Family traits) inherited cancer for immigrants separated from family history*" (CS128)

*Subtheme 1.4 Prevention.* Several participants cited the need for research into cancer prevention, although no details were provided (*"preventing it in the first place"* (CS129)). Not only prevention of the primary cancer, but also prevention of cancer recurrence was mentioned (*"alleviate recurrence"* (CS176)).

*Subtheme 1.5 Rare cancers.* Participants mentioned the need for research into rare cancers, often in relationship to their own diagnosis or that of their loved one. A need for more information on these unusual cancer types was expressed (*"Research into less common cancers—getting more awareness"* (CS104)).

## Theme 2. Can you detect cancer earlier and treat it better?

*Subtheme 2.1 Screening and early detection.* This was an area of high importance to many participants. Some participants felt that their general practitioners missed the early signs of cancer

and that the required testing was not done, leading to late diagnosis and poor prognosis. Hence, a *"more consistent screening programme"* and *"new methods for early detection"* (*CS116*) were listed. In addition, *"painless testing"* (*CS177*), was thought important.

> *"More. GP [general practitioner] & A&E [accident and emergency] testing before too late for diagnosis" (CS199)*

> *"Leaving the ambulance at the bottom of the cliff will never be the best method; Screening early is the best method" (CS071)*

As cancer incidence increases with age, most screening programs have a minimum age cut-off. However, some cancers are missed in those who are defined too young for screening, which was a concern for participants.

> *"I am sure the researchers are doing a fantastic job with all their researching but I do think that a lot of screening like bowel and breast cancer and many others, the age to test should be brought in for the younger generations as well" (CS036)*

*Subtheme 2.2 Complementary and holistic treatments.* Over a quarter of Māori participants mentioned this subtheme, demonstrating their wish for further information and research in this area. Alternative and complementary treatments mentioned included *"natural medicines"* (CS177), *"herbal treatments"* and *"cannabis"* (*CS173*), *"exercise"* (*CS104*) and *"music"* (*CS187*). Spirituality and holistic therapy were important themes.

> *"Complementary and alternative therapies, cannabis and vitamin C along with Māori Health and rongoā treatments" (CS020)*

> *"Hospitals require to have facilities that offer holistic environments that ease, calm, relax, prepare patients physically, mentally, spiritually" (CS059)*

A need for additional information on, and access to, alternative or complementary therapies was apparent. Clear information on effectiveness and availability of all potential therapies was linked to trust in the treating physician.

> *"Trust & education on what is available for alternative medication for our tūroro" (CS197)*

> *"For doctors and treatment to take into account faith and an holistic approach to treatment. To be able to be open minded and support a patient's right to take the journey however they see fit. To look at the use of alternative therapies and diet" (CS064)*

*Subtheme 2.3 New drugs and treatments.* Many participants mentioned the need for better or *"painless treatment"* (*CS177*), as well as a need for treatments with fewer side effects.

> *"How to reduce side effects of chemotherapy/nausea" (CS016)*

The need for more research into specific new drugs included *"immunotherapy"* (*CS004*) and *"Keytruda"* (*CS015*). Participants expressed a need for *"more drug trials"* (*CS011*) and *"development of new cancer drugs"* (*CS119*).

*Subtheme 2.4 International standards and collaborations.* Participants wanted world-class treatment and were concerned that some in Aotearoa might be missing out. For this reason, participants supported greater collaboration with other international researchers and

clinicians, suggesting that this would bring expertise and access to the latest treatments to Aotearoa.

"*Access to clinical trials from all NZ centres—international/Australian trials*" (CS104)

"*More research and communicate to other area, cities and countries who have more experience about it, more successful cases that they have done in past*" (CS086)

### Theme 3. Can you look after our patients better?

The health delivery system was an important theme, and whether it met the needs of the people it means to serve. This theme in particular showed the crossover between research wanted and support sought.

*Subtheme 3.1 Role of health provider.* Information provided during clinical visits is often overwhelming. Participants commented on the approach of their general practitioner or oncologist to delivering the diagnosis. A common concern was difficulty in understanding medical terminology and a lack of the right tools for this.

"*The way we are told the results when the bad news is delivered*" (CS178)"*Layman's terms if the person does not understand; fully understood; repeat if in doubt*" (CS018)

Participants at times felt lost in the system, some describing it as chaotic, with a lack of continuity between services. Participants reported finding it hard to find support.

"*The process once cancer is diagnosed—I believe the process is flawed and causes delays. . . it can feel like there are too many involved and that treatment of the patient becomes a bit like being on a production line*" (CS136).

*Subtheme 3.2 Access to care & finances & amenities.* Finances were a key concern for participants ("*more national health assistance*" (CS046)), which is likely linked to time away from work or losing their job. Another major concern was the effect of cancer diagnosis and treatment on whānau/family, and how they were going to cope. The practical issues and daily stresses experienced by patients with cancer included (a perceived) lack of usable Wi-Fi (for Zoom communication with whānau) (CS189), public toilets (CS179) and parking during appointments (CS204).

"*Transport sorted out for family to & from hospital (can be very costly for low income families) eg passes for family & support people*" (CS180).

*Subtheme 3.3 Public education and cancer information.* Participants expressed the view that education and information should be more accessible, and that more resources for those diagnosed with cancer were needed in the community.

"***A more comprehensive website or information so you don't have to search** everywhere for information. A helpline that follows up once diagnosed, to give you support and talk through options, gives information.*" (CS166)

*Subtheme 3.4 Emotional burden of cancer & survivorship & peer support.* This subtheme was only mentioned by women, who felt that additional support and mentorship would help those living with cancer. It was felt that looking after tūroro at home was often challenging.

*"Stress relief, for everyone concerned around family"* (CS179)

*"Emotional side of dealing with cancer and also what it does for families. Especially the spouses of those dealing with cancer. I feel they don't want to understand its harder for them to deal with"* (CS167)

*Subtheme 3.5 Culturally sensitive approaches.* Some participants said they preferred to see 'someone like me', with regards to ethnicity, age, spirituality or religion, raising the requirement of clinical staff to be culturally aware and competent. This was not specific to Māori but included other ethnicities.

*"Doctors/staff/nurses etc to be more informed/open to and with Māori health practices, ethnic background"* (CS058)

*"Information and health promotions focuses "pakeha" and it comes over as only pakeha get bowel cancer therefore many ethnic people don't know symptoms of bowel cancer such as bowel habit changes"* (CS149)

## Quantitative assessment of free-text entries

To get further insight into participant's priorities, we also quantified the free-text entries. Overall, 52% of participants provided free-text answers, with 85% of Māori and 44% of non-Māori participants. Of the 33 Māori participants who provided free-text input, we identified 60 concepts (**Fig 1**). Māori participants commented most often on *Complementary and holistic treatments* (n = 10, subtheme 2.2) and *Access to care & finances & amenities* (n = 8, subtheme 3.2), followed by *Screening and Early detection* (n = 6, subtheme 2.1) and *Culturally sensitive approaches* (n = 6, subtheme 3.5). Of the 73 non-Māori participants who provided free-text input, we identified 124 concepts (**Fig 1**). Non-Māori participants wrote most often about

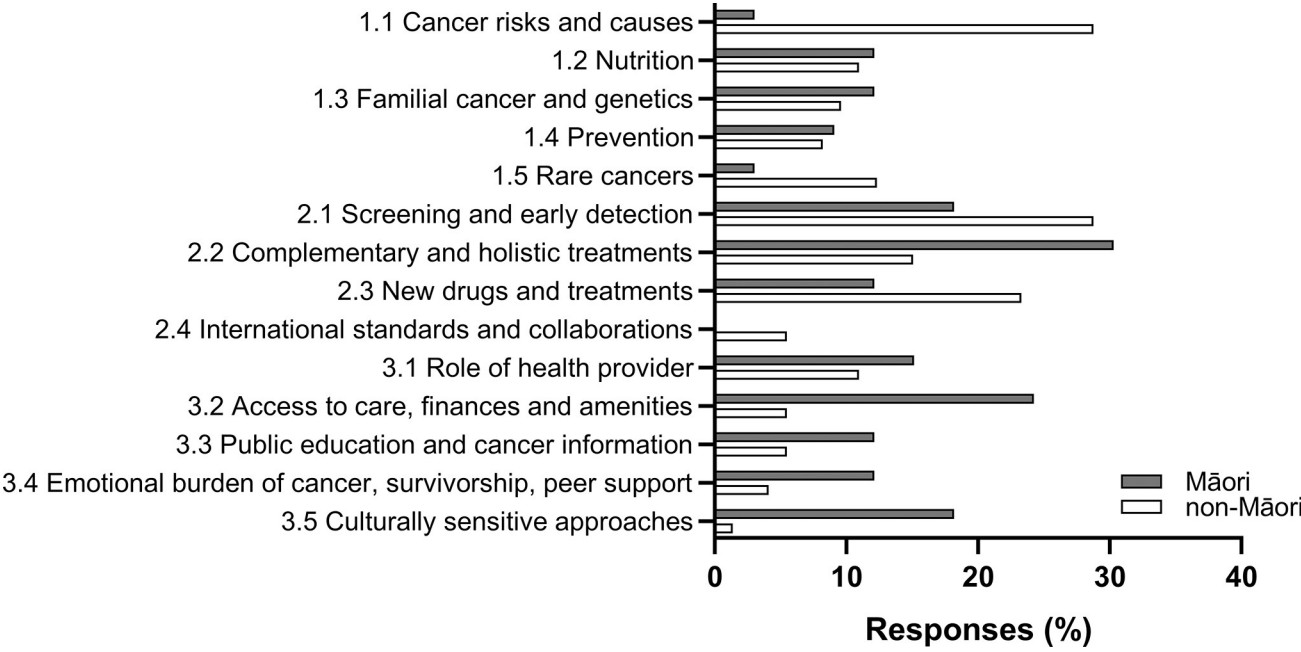

**Fig 1. Thematic answers according to participants' ethnicity.**

*Cancer risks and causes* (n = 21, subtheme 1.1) and *Screening and early detection* (n = 21, subtheme 2.1), followed by *New drugs and treatments* (n = 17, subtheme 2.3).

Participants' free-text entries were analysed by inductive thematic analysis and coding methods, with 3 broad themes and multiple subthemes identified. The proportion of responses under each subtheme are shown. Of 205 participants, n = 106 participants provided comments, including 33 who identified as Māori (= 100%) and 73 who identified as non-Māori (= 100%).

Of the 129 tūroro/patients, 48% completed free-text, compared to 59% of whānau/family. Tūroro most often commented on *Early detection* (n = 17, subtheme 2.1), followed by *Complementary therapy* (n = 14, subtheme 2.2) and *New drugs* (n = 13, subtheme 2.3). Whānau commented equally often on *Cancer risks* (n = 10, subtheme 1.1) and *Early detection* (n = 10, subtheme 2.1).

Of the 123 wahine/females, 56% completed free-text, whereas 46% of tāne/males did. Wahine commented most often on *Early detection* (n = 16, subtheme 2.1) followed by *Cancer risks* (n = 15, subtheme 1.1), then *Complementary therapy* (n = 14, subtheme 2.2). Tāne wrote about *Early detection* (n = 11, subtheme 2.1) most often, followed equally by *Cancer risks* (n = 7, subtheme 1.1), *Complementary therapy* (n = 7, subtheme 2.1) and *New drugs* (n = 7, subtheme 2.3).

## Discussion

This study aimed to gain an understanding of research priorities from the perspective of those with a cancer diagnosis or their whānau/family in Aotearoa/New Zealand. Emphasis was placed on obtaining views from diverse participants, specifically from Māori, but also from tūroro/patients and whānau/family, and wahine/females and tāne/males. A close collaboration with Māori health workers enabled recruitment of over twice the proportion of Māori participants compared to the general population in Canterbury. The use of Te Reo Māori in parts of the survey and in reporting these results is done on purpose to enhance inclusivity of Aotearoa's indigenous population.

The survey used several approaches, asking participants to rate the priorities of broad research areas, to select priorities from a list of cancer-related topics, and finally to expand on or provide free text comments on research priorities not already covered. Virtually all participants viewed cancer research as a very important mission, with largely similar priorities between Māori and non-Māori participants. Keywords 'Cancer screening', 'Quality of Life' and 'Development of new drugs' were chosen most often. Free-text analysis identified three themes; *'Genetics and Prevention'* (understanding environmental, genetic risks), *'Early Detection and Treatment'* (improving screening, treatments), and *'Service Delivery'* (information and support, culturally sensitive treatment).

Health disparities have been reported worldwide for indigenous and minority populations [16, 17]. In addition, it has been recognised that much of the research to date has lacked ethnic and other diversity, and there have been calls for a change in recruitment practices to ensure equitable participation [18]. It is vital to ensure appropriate reporting of health research involving indigenous people and not to, inadvertently, propagate bias [19]. For this reason, the CONSIDER statement has been developed, which is a framework for researchers that aims to 'strengthen research practices and reporting' and 'to support indigenous health equity' [18]. Our study has attempted to address these issues and follow the CONSIDER guidelines by consulting with Māori advisors from the start, by working in partnership with Māori throughout including recruitment, data analysis and interpretation, and by including te reo Māori in participant's narratives and tikanga Māori processes of engagement. In addition, three of our contributing authors identify as Māori.

The majority of participants questioned why they had cancer, indicating that they supported more research on aetiology, and related to that, prevention. Participants were

concerned about the risks of (rare) heritable cancers. Attitudes to genetic testing, when mentioned, were positive, with examples of local celebrities (eg the musician Stan Walker) used to illustrate the utility of testing for whānau. Genetic testing is still seen as potentially contentious for Māori, due to genetic samples and whakapapa being taonga that don't just belong to the individual but have implications for whānau and iwi [20]. The absence of negative attitudes to genetic testing in this sample may be because they were facing life-threatening illness themselves with potential implications for wider whānau and mokopuna (descendent), hence the risk-benefit balance may be different for individuals in those circumstances.

Although we identified strong lived experience support for ongoing research into cancer screening and early diagnosis, it is recognised that screening, in particular, is complex and comes with its own controversies. These issues include false positive and false negative findings, potential harm from the test itself (e.g. radiation exposure), the possibility of overdiagnosis and how to handle incidental findings [21]. A current issue being debated in Aotearoa is the starting age of bowel screening, particularly as, in comparison to non-Māori, a greater percentage of bowel cancers in Māori occur before the age of 60 years, the current age of screening [22]. It is clear that further research is needed into availability of screening and improved early identification methods.

The areas of smoking and alcohol were seldom chosen as research priorities. This appears at odds with the high rates of smoking and alcohol (ab)use in Aotearoa [3]. The cancers most closely related to tobacco smoking and alcohol show high rates; in 2018 lung cancer had the highest cancer mortality rate (21/100,000) and liver cancer was also relatively high (3.4/100,000) [1, 2]. Some possible explanations for this apparent lack of interest may be that these behaviours are deemed to be already well understood as risk factors, or that stigma and blame surrounds these behaviours, or that it is too late for those who have the disease.

Quite apart from stigma towards cancers related to tobacco or alcohol-use amongst those affected by cancer, this stigma is also said to influence research in this area. Kamath [23] reports that this stigma hinders research funding applications, whereas some research areas with lower societal cancer burden, such as rare cancers, apparently tend to receive disproportionate attention. These issues represent an equity issue in Aotearoa.

It is of interest that only 20% of participants endorsed "equity" as a research priority; the lowest ranking of all priorities. It appears that this heading was not recognised by the participants. From free-text answers it seemed that equity was indeed of high priority but was placed under other guises. Indeed, many of the research areas mentioned (eg screening) and issues raised (eg service delivery) directly reflect equity matters. In future research, consideration should be given to elaborating on the meaning of the word to see whether that changes the current low ranking by those with lived experience, which is at odds with the priority placed on equity (in research) by researchers and research funders [24].

Participants were interested in incorporation of complementary or alternative treatments. There is increasing recognition of the benefits of incorporating holistic treatments alongside medical treatments for cancer [25–27]. Complementary medicine strategies are of interest in Aotearoa [28] and may improve quality of life of patients with cancer, but robust clinical data is sparse and the value for our population is unknown. The need for complementary/alternative therapies may also reflect a mistrust of modern medicine or the health services, or reflect a structural barrier in terms of how the health system has been designed [6].

Several participants felt strongly about spirituality and religion, which are well-known coping mechanisms for patients with cancer [29]. Spirituality recognises a sense of something greater than oneself, and can be viewed as separate to religion, or intimately linked. Specifically, Māori origins of spirituality are deeply embedded within their genealogy and ancestral heritage (whakapapa) [30], and should therefore be addressed as part of cancer support.

Consistent with this and the call for culturally sensitive treatment, there is increasing interest in researching the benefits of traditional Rongoā Māori practices to improve wellbeing for Māori patients [30]. It is important to note that Rongoā Māori is more than another complementary healing medicine, and is recognised as a holistic life practice, embedded within a Te Ao Māori (Māori world) cultural values framework [30]. Rongoā Māori is protected under Te Tiriti o Waitangi (Aotearoa's founding document) and is recognised by the NZ Māori Health Authority (Te Aka Whai Ora) [31]. The Rongoā Māori work programme identifies funding pathways supported by the NZ Health Research Council [31].

Although the focus of this study was on cancer research priorities, a number of participants commented on areas of dissatisfaction with service delivery and current practises. The comments (indirectly) addressed equity as the majority of the remarks reflected inequities related to ethnicity, socio-economic, education and health literacy. Our survey showed the value of an independent party talking directly with tūroro and whānau, such that they felt empowered to report problems and areas for improvement. Consistent with our findings, many previous studies have noted the complex issues involved in the initial communication and receipt of a cancer diagnosis, and the fact that understanding emerges over time, so the provision of information whenever the person is ready and needs it, is important [32–34]. Information on cancer produced specifically for different ethnicities or age groups could also be helpful.

The survey also providing the opportunity to comment on the need for practical support, such as financial aid, Wifi and parking. A cancer diagnosis negatively affects the financial wellbeing of most tūroro and their whānau due to treatment-related costs and loss of income, and varies by socioeconomic status [35]. These additional stresses make the cancer journey harder and impact on patients' survival [36, 37].

The strengths of this study include that, to our knowledge, it is the first study in Aotearoa identifying the views of tūroro and their whānau on priorities for cancer research. The partnership with Māori health workers contributed to the high participation of Māori in this study and the detailed comments provided.

There are a number of limitations of this study. First, although free text answers provided valuable additional insight, it is also clear that the (limited) words written down do not fully reflect the participants suggestions or emotions. Future research should include qualitative interviews which would allow more in-depth understanding of participants comments. A hui (gathering) process to gather Māori-rich information would be another vital avenue to consider. Another limitation is that the survey, being carried out at a major tertiary hospital, may not have captured priorities and aspirations for whānau living rurally.

In conclusion, cancer research was a high priority for those living with cancer, with support for research across the spectrum from aetiology and prevention, to better, more effective cancer treatments. It has become obvious, though, that many participants request that we, as researchers, listen to tūroro/patients and whānau/family regarding their immediate and practical needs. From their comments, there is clearly a need for further research into areas of treatment satisfaction and unmet needs in service delivery, including the need for better information, culturally appropriate treatment and incorporation of complementary and Rongoā Māori practices to improve quality of life while living with cancer. These findings will inform future cancer research in Aotearoa.

## Supporting information

**S1 Data.**
(XLSX)

## Author Contributions

**Conceptualization:** Karen Keelan, Bridget A. Robinson, Gabi U. Dachs.

**Data curation:** Millie de Vries, Gabi U. Dachs.

**Formal analysis:** Jennifer Jordan, Gabi U. Dachs.

**Funding acquisition:** Bridget A. Robinson, Gabi U. Dachs.

**Investigation:** Millie de Vries, Tiria Stewart.

**Supervision:** Tiria Stewart, Theona Ireton, Karen Keelan, Gabi U. Dachs.

**Writing – original draft:** Gabi U. Dachs.

**Writing – review & editing:** Millie de Vries, Tiria Stewart, Theona Ireton, Karen Keelan, Jennifer Jordan, Bridget A. Robinson, Gabi U. Dachs.

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
