## [Decision Letter · Decision Letter 0]

26 May 2023

PONE-D-23-10085Patients’ and carers’ priorities for cancer research in Aotearoa/New ZealandPLOS ONE

Dear Dr. Dachs,

Thank you for submitting your manuscript to PLOS ONE. After careful consideration, we feel that it has merit but does not fully meet PLOS ONE’s publication criteria as it currently stands. Therefore, we invite you to submit a revised version of the manuscript that addresses the points raised during the review process.

 or perceived impact.

In particular, make sure to revise your statement of a "mixed-method approach". Mixed method usually refers to a study design combining quantitative and qualitative data. The inclusion of a few free-text questions in a survey does not qualify for qualitative research.It would also be helpful to put the translation of Maori expressions in an alphabetical order to facilitate legibility.Further comments by our reviewers can be found below. Please submit your revised manuscript by Jul 10 2023 11:59PM. If you will need more time than this to complete your revisions, please reply to this message or contact the journal office at plosone@plos.org. Please include the following items when submitting your revised manuscript:A rebuttal letter that responds to each point raised by the academic editor and reviewer(s). You should upload this letter as a separate file labeled 'Response to Reviewers'.A marked-up copy of your manuscript that highlights changes made to the original version. You should upload this as a separate file labeled 'Revised Manuscript with Track Changes'.An unmarked version of your revised paper without tracked changes. You should upload this as a separate file labeled 'Manuscript'.

We look forward to receiving your revised manuscript.

Kind regards,

Thomas Behrens

Academic Editor

PLOS ONE

Reviewers' comments:

Reviewer's Responses to Questions

**Comments to the Author**

1. Is the manuscript technically sound, and do the data support the conclusions?

Reviewer #1: Partly

Reviewer #2: Yes

2. Has the statistical analysis been performed appropriately and rigorously? 

Reviewer #1: No

Reviewer #2: Yes

3. Have the authors made all data underlying the findings in their manuscript fully available?

Reviewer #1: No

Reviewer #2: No

4. Is the manuscript presented in an intelligible fashion and written in standard English?

Reviewer #1: Yes

Reviewer #2: Yes

5. Review Comments to the Author

Reviewer #1: Dear Editor,

Thank you for the invitation to review the manuscript titled: Patients’ and carers’ priorities for cancer research in Aotearoa/New Zealand. The authors had clearly identified a knowledge gap in cancer research with an aim to improve care for cancer patients living in Aotearoa New Zealand.

Please see attached comments included throughout the manuscript for the authors to consider carefully for improvement of the overall manuscript. One of the key comments is about the chosen study design. The author has stated this is a mixed methods study. An MMR includes both quantitative and qualitative study designs. Based on the description here, this is a quantitative study (cross-sectional study) with a (few) open-ended question(s) for additional inputs. Suggest for the author to have a clear justification for how they have chosen MMR as the appropriate study design. Generally, the manuscript flows coherently, with some sections/sentences need more elaboration to clarity.

Reviewer #2: This is a well-written manuscript on a crucial topic aiming at describing the priorities of cancer patients for cancer research in Aotearoa New Zealand. Some results might however need to be reported or presented differently to get a better understanding of the data.

Here are a few specific comments and suggestions, mostly on the quantitative analyses:

General comment: The first question is analysed for all participants only while the other two questions are also analysed by ethnic group (Māori vs. non-Māori), by gender and by tūroro/whanau. It would be great to report the stratified results also for the first question. It would also be great to report figures by gender and by tūroro/whanau in tables/figures (either in the manuscript or in appendices).

Page 4, third paragraph: Some words seem to be missing after “Research priorities”.

Page 8, “Rating research areas” section: “Improving Cancer Outcomes” is the research area with the highest percentage of high/very high ratings. However, Figure 1 doesn’t show that it is “marginally higher priority” than others, i.e., the percentage of high/very high ratings for “Cancer prevention” and “side effects” seems to be over 95%, so probably only 1-2% lower than the percentage of high/very high ratings for “Improving Cancer Outcomes”. On the other hand, the percentage of high/very high ratings for diagnosis/prognosis is about 90%, which is lower than the other three research areas. It is hard, however, to see the exact percentages on the graph and it would be useful to add them on. A table with n (%) might be more appropriate and statistical testing might be used to determine whether there is any significant difference of rating between research areas.

Page 8, “Cancer-related keywords” section: It is interesting to know the top 10 chosen words for Māori and non-Māori participants separately, but it would also be nice to see the difference between groups for each key word. For example, it would be nice to know the percentage of Māori participants who chose “pain relief and cancer” and how much it differs from the percentage of non-Māori who chose this keyword.

Page 9, “Cancer-related keywords” section: Results differences between tūroro and whanau and between genders are interesting and it would be great to have them reported in Table 2 or appendices.

Table 1: Even though non-Māori participants have been grouped together for the analyses, it would be interesting to see the breakdown by ethnicity in this table.

Figure 2: I think these results should be presented differently. The comparison of number of responses between Māori and non-Māori participants does not mean much here as 33 Māori participants vs. 73 non-Māori participants provided free-text input. Percentages should be reported to allow the comparison. A table with n (%) might be easier to interpret.

6. PLOS authors have the option to publish the peer review history of their article (what does this mean?). If published, this will include your full peer review and any attached files.

Reviewer #1: No

Reviewer #2: No

---

## [Author Response · Author response to Decision Letter 0]

5 Jul 2023

We have carefully responded to every suggestion and critique by the editor and the two reviewers, and have modified the manuscript accordingly (all details also in Response to Reviewers file).

Point-by-point response to the reviewers’ comments.

Editor: In particular, make sure to revise your statement of a "mixed-method approach". Mixed method usually refers to a study design combining quantitative and qualitative data. The inclusion of a few free-text questions in a survey does not qualify for qualitative research.

The term ‘mixed methods’ was removed from the manuscript, and reworded in Methods (p7) and Discussion (p20).

It would also be helpful to put the translation of Maori expressions in an alphabetical order to facilitate legibility.

Māori terms and translation are now ordered alphabetically (p3).

In your Data Availability statement, you have not specified where the minimal data set underlying the results described in your manuscript can be found.

Minimal data has been added as Supplementary Table 1.

Further comments by our reviewers can be found below.

Reviewer #1: Dear Editor,

Thank you for the invitation to review the manuscript titled: Patients’ and carers’ priorities for cancer research in Aotearoa/New Zealand. The authors had clearly identified a knowledge gap in cancer research with an aim to improve care for cancer patients living in Aotearoa New Zealand.

Please see attached comments included throughout the manuscript for the authors to consider carefully for improvement of the overall manuscript. One of the key comments is about the chosen study design. The author has stated this is a mixed methods study. An MMR includes both quantitative and qualitative study designs. Based on the description here, this is a quantitative study (cross-sectional study) with a (few) open-ended question(s) for additional inputs. Suggest for the author to have a clear justification for how they have chosen MMR as the appropriate study design. Generally, the manuscript flows coherently, with some sections/sentences need more elaboration to clarity.

The Data analysis section in Methods (p7) was reworded in response: ‘This is a quantitative, cross-sectional study with an open-ended question for additional input.’

Additional comments throughout manuscript from marked pdf:

what is the purpose of comparison for this project?

Prioritised ethnicity classification was used in order to identify specific Māori priorities. We therefore added: ‘This is a high priority for health and research organisations in Aotearoa/NZ to address health inequities.’ (p6)

Some description the method of compilation of these keywords is desired.

Additional information was added: ‘In section 3, participants were provided with a list of 44 cancer-related keywords, compiled from discussions with scientists and clinicians at the University of Otago Christchurch and Christchurch Hospital.’ (p7)

It is unclear if any statistical test was completed for Maaori and non-Maaori (ref first paragraph, last sentence)

The following was added to Methods: ‘Normality of data was tested using the D'Agostino & Pearson test, followed by the Mann Whitney t-test (for non-parametric analysis) to compare between groups, with p < 0.05 being statistically significant.’ (p7)

mixed methods study includes both quantitative and qualitative study designs. Based on the description here, this is a quantitative study (cross-sectional study) with a (few) open ended question(s) for additional inputs.

The term ‘mixed methods’ was removed from the manuscript. The Data analysis section in Methods was reworded in response: ‘This is a quantitative, cross-sectional study with an open-ended question for additional input.’ (p7)

Page 11, second last sentence: this sentence is ambiguous; also be cautious to generalise findings from a few quotes/code with 'certainty' ('.the need.... was clear.')

The wording was rephrased to clarify: ‘A need for additional information on, and access to, alternative or complementary therapies was apparent. Clear information on effectiveness and availability of all potential therapies was linked to trust in the treating physician.’ (p16)

Discussion, Page 15, first para, last sentence. This needs to be included in the introduction section as well.

We added this to the end of the background section: ‘We purposefully used Te Reo Māori in parts of the survey to enhance participation and improve inclusivity of Aotearoa’s indigenous population.’ (p5)

Page 16, para 3: I would place this in the intro and method sections; "the CONSIDER statement..." in the method section under recruitment approach. The discussion section to discuss the results/findings.

We chose to keep the discussion of the CONSIDER statement in the Discussion, but have expanded it to emphasise how it was addressed in our study: ‘Our study has attempted to address these issues and follow the CONSIDER guidelines by consulting with Māori advisors from the start, by working in partnership with Māori throughout including recruitment, data analysis and interpretation, and by including te reo Māori in participant’s narratives and tikanga Māori processes of engagement. In addition, three of our contributing authors identify as Māori.’ (p20/21) 

page 18, last para, last sentence. Please be more elaborative how these two government agencies providing support for Rongoā Māori

Additional explanations have been added: ‘Rongoā Māori is protected under Te Tiriti o Waitangi (Aotearoa’s founding document) and is recognised by the NZ Māori Health Authority (Te Aka Whai Ora) 31. The Rongoā Māori work programme identifies funding pathways supported by the NZ Health Research Council 31.’ (p22/23)

Page 19 - what do you mean by independent party? who are they? are specially trained to communicate with cancer patients? More description in the methods section would be beneficial.

The following was added to methods: ‘The two recruiters are specially trained to communicate with patients and were independent from the tūroro/patient’s treatment team.’ (p6)

Tables and figures: I was expecting to see the results by patients and carer groups (as per the title)

The title of Table 1 was modified accordingly: ‘Table 1: Characteristics of the cohort.’

Figure2: reporting the proportion is more meaningful

The figure (now Figure 1) has been modified accordingly.

Reviewer #2: This is a well-written manuscript on a crucial topic aiming at describing the priorities of cancer patients for cancer research in Aotearoa New Zealand. Some results might however need to be reported or presented differently to get a better understanding of the data.

Here are a few specific comments and suggestions, mostly on the quantitative analyses:

General comment: The first question is analysed for all participants only while the other two questions are also analysed by ethnic group (Māori vs. non-Māori), by gender and by tūroro/whanau. It would be great to report the stratified results also for the first question. It would also be great to report figures by gender and by tūroro/whanau in tables/figures (either in the manuscript or in appendices).

A new table (Table 2) was created, showing more details of research priorities by subgroup. An expanded Table 3 was created showing keyword choices according to ethnicity, sex and status, as well as keywords across the entire cohort.

Page 4, third paragraph: Some words seem to be missing after “Research priorities”.

This has been corrected; ‘of’ was removed.

Page 8, “Rating research areas” section: “Improving Cancer Outcomes” is the research area with the highest percentage of high/very high ratings. However, Figure 1 doesn’t show that it is “marginally higher priority” than others, i.e., the percentage of high/very high ratings for “Cancer prevention” and “side effects” seems to be over 95%, so probably only 1-2% lower than the percentage of high/very high ratings for “Improving Cancer Outcomes”. On the other hand, the percentage of high/very high ratings for diagnosis/prognosis is about 90%, which is lower than the other three research areas. It is hard, however, to see the exact percentages on the graph and it would be useful to add them on. A table with n (%) might be more appropriate and statistical testing might be used to determine whether there is any significant difference of rating between research areas.

The new Table 2 shows details of the cancer research priorities for all participants as well as by subgroup. Original Figure 1 was removed and the text was altered: ‘The four research areas were ranked in order of Improving Cancer Outcomes, Diagnosis/Prognosis of Cancer, Cancer Prevention and Reducing Side Effects. When analysed by sub-groups, tūroro rated Side Effects and Outcomes significantly lower than whānau, whereas wahine rated all four areas significantly higher than tāne (Table 2).’ (p10)

Page 8, “Cancer-related keywords” section: It is interesting to know the top 10 chosen words for Māori and non-Māori participants separately, but it would also be nice to see the difference between groups for each key word. For example, it would be nice to know the percentage of Māori participants who chose “pain relief and cancer” and how much it differs from the percentage of non-Māori who chose this keyword.

Table 3 was modified and now provides details of the keyword choices according to ethnicity, sex and status, as well as those for the entire cohort. All keywords chosen by >50% of at least one subgroup are shown.

Page 9, “Cancer-related keywords” section: Results differences between tūroro and whanau and between genders are interesting and it would be great to have them reported in Table 2 or appendices.

An expanded Table 3 provides details of the keyword choices according to ethnicity, sex and status. 

Table 1: Even though non-Māori participants have been grouped together for the analyses, it would be interesting to see the breakdown by ethnicity in this table.

Additional ethnicity details were added to Table 1.

Figure 2: I think these results should be presented differently. The comparison of number of responses between Māori and non-Māori participants does not mean much here as 33 Māori participants vs. 73 non-Māori participants provided free-text input. Percentages should be reported to allow the comparison. A table with n (%) might be easier to interpret.

The figure (New Figure 1) has been modified to show the percentage responses for the two ethnic groups.

---

## [Editor Report · Decision Letter 1]

7 Aug 2023

Patients’ and carers’ priorities for cancer research in Aotearoa/New Zealand

PONE-D-23-10085R1

Dear Dr. Dachs,

We’re pleased to inform you that your manuscript has been judged scientifically suitable for publication and will be formally accepted for publication once it meets all outstanding technical requirements.

Kind regards,

Thomas Behrens

Academic Editor

PLOS ONE

---

## [Editor Report · Acceptance letter]

11 Aug 2023

PONE-D-23-10085R1 

Patients’ and carers’ priorities for cancer research in Aotearoa/New Zealand 

Dear Dr. Dachs:

I'm pleased to inform you that your manuscript has been deemed suitable for publication in PLOS ONE. Congratulations! Your manuscript is now with our production department. 

Kind regards, 

on behalf of

Prof. Thomas Behrens 

Academic Editor

PLOS ONE